# Antitumor Effect of Cabozantinib in Bone Metastatic Models of Renal Cell Carcinoma

**DOI:** 10.3390/biology10080781

**Published:** 2021-08-16

**Authors:** Michele Iuliani, Sonia Simonetti, Francesco Pantano, Giulia Ribelli, Alberto Di Martino, Vincenzo Denaro, Bruno Vincenzi, Antonio Russo, Giuseppe Tonini, Daniele Santini

**Affiliations:** 1Department of Medical Oncology, Campus Bio-Medico University of Rome, 00128 Rome, Italy; m.iuliani@unicampus.it (M.I.); s.simonetti@unicampus.it (S.S.); g.ribelli@unicampus.it (G.R.); b.vincenzi@unicampus.it (B.V.); g.tonini@unicampus.it (G.T.); d.santini@unicampus.it (D.S.); 2Department of Biomedical and Neurimotor Sciences (DIBINEM), 1st Orthopaedic Clinic, IRCCS Istituto Ortopedico Rizzoli, 40126 Bologna, Italy; alberto.dimartino@ior.it; 3Department of Orthopaedic and Trauma Surgery, Campus Bio-Medico University of Rome, 00128 Rome, Italy; v.denaro@unicampus.it; 4Section of Medical Oncology, Department of Surgical, Oncological and Oral Sciences, University of Palermo, 90133 Palermo, Italy; antonio.russo@usa.net

**Keywords:** cabozantinib, renal cell carcinoma, osteoblasts

## Abstract

**Simple Summary:**

Bone metastasis is a common and devastating feature of advanced renal cancer. Skeletal metastases are very destructive in patients with renal cell carcinoma (RCC), compromising the bone integrity and leading to fractures, pain, nerve compressions and hypercalcemia, with a negative impact on patient survival. Cabozantinib is a receptor tyrosine kinase inhibitor approved for the treatment of advanced RCC patients. Recently, preclinical studies demonstrated that cabozantinib is able to also modulate bone cell activity. To understand whether and how the antitumor activity of cabozantinib could be influenced by the bone microenvironment, in vitro co-culture models of renal cancer cells and osteoblasts (OBs) were developed to reproduce the bidirectional interplay between tumor cells and bone. The data showed that cabozantinib preserves its efficacy in these coculture models and exerts an additional indirect antitumor activity mediated by OBs. Indeed, cabozantinib is able to inhibit specific OB proliferative signals that, in turn, could affect RCC cell growth. To elucidate how OBs stimulate RCC cell growth could be useful in designing novel and more effective anticancer strategies to improve the efficacy of the existing treatments.

**Abstract:**

Background: The presence of bone metastases in renal cell carcinoma (RCC) negatively affects patients’ survival. Data from clinical trials has highlighted a significant benefit of cabozantinib in bone metastatic RCC patients. Here, we evaluated the antitumor effect of cabozantinib in coculture models of renal cell carcinoma (RCC) and osteoblasts (OBs) to investigate whether and how its antiproliferative activity is influenced by OBs. Methods: Bone/RCC models were generated, coculturing green fluorescent protein (GFP)-tagged Caki-1 and 786-O cells with human primary OBs in a “cell–cell contact” system. RCC proliferation and the OB molecular profile were evaluated after the cabozantinib treatment. Results: The Caki-1 cell proliferation increased in the presence of OBs (*p* < 0.0001), while the 786-O cell growth did not change in the coculture with the OBs. The cabozantinib treatment reduced the proliferation of both the Caki-1 (*p* < 0.0001) and 786-O (*p* = 0.03) cells cocultured with OBs. Intriguingly, the inhibitory potency of cabozantinib was higher when Caki-1 cells grew in presence of OBs compared to a monoculture (*p* < 0.001), and this was similar in 786-O cells alone or cocultured with OBs. Moreover, the OB pretreatment with cabozantinib “indirectly” inhibited Caki-1 cell proliferation (*p* = 0.040) without affecting 786-O cell growth. Finally, we found that cabozantinib was able to modulate the OB gene and molecular profile inhibiting specific proliferative signals that, in turn, could affect RCC cell growth. Conclusions: Overall, the “direct” effect of cabozantinib on OBs “indirectly” increased its antitumor activity in metastatic RCC Caki-1 cells but not in the primary 786-O model.

## 1. Introduction

Renal cell carcinomas (RCC) comprise a heterogeneous group of malignant neoplasms, including clear cell, papillary and chromophobe histological subtypes. The most common sites of metastatic involvement are the lung, lymph nodes, bone, liver, adrenal and brain [1]. Moreover, isolated pancreatic metastasis can develop in RCC patients typically a long time after nephrectomy [2]. Bone metastases occur in 20–35% of RCC patients, leading to mainly osteolytic lesions that compromise the bone integrity and patient outcomes [3]. In particular, the presence of bone metastases in RCC patients negatively affects the progression-free survival (PFS) and overall survival (OS) [3]. Moreover, bone metastases are associated with skeletal-related events (SREs), including pain, impending fractures, nerve compressions, hypercalcemia and even pathological fractures, which may require surgical interventions and other therapy [4,5]. Data about the benefit of tyrosine kinase inhibitors (TKIs), commonly used in the treatment of advanced RCC, on the survival of patients with bone metastasis remain unclear.

A recent phase III trial (METEOR) showed that cabozantinib, an orally bioavailable receptor tyrosine kinase (RTK) inhibitor with potent activity against multiple RTKs (MET, RET, VEGFR2, FLT3 and c-KIT) and Tyro3, Axl and Mertk (TAM) RTKs [6] had a significant clinical benefit for bone metastatic RCC patients [7]. In particular, in a prespecified subgroup analysis of RCC patients with bone metastasis, a marked prolongation PFS was observed in the cabozantinib arm compared to the everolimus group (7.4 months vs. 2.7 months). In addition, the incidence of subsequent SREs was reported in 16% of patients treated with cabozantinib vs. 34% in the everolimus arm [7]. However, some infrequent but serious events have been reported, especially in the later phases of treatment, including hemorrhagic and thrombotic events. In this regard, the administration of oral anticoagulants for cabozantinib-associated thrombosis could be useful in clinical practice to reduce the morbidity and mortality of these patients [8].

Cabozantinib is approved in advanced RCC patients previously treated with VEGF inhibitors [9] and as the first line in treatment-naïve patients of intermediate or poor risks [10]. Recently, the USA Food and Drug Administration (US FDA) extended its indication to all advanced RCC patients.

In the last years, our research group demonstrated that primary osteoclasts and osteoblasts (OBs) express the specific targets of cabozantinib, and its administration can inhibit osteoclast differentiation and bone resorption activity [11]. Another study published by our team demonstrated that cabozantinib could be a new potential treatment against osteosarcoma, targeting both tumors and their microenvironments [12]. Starting from this evidence, the aim of the study is to evaluate the antitumor effect of cabozantinib in in vitro coculture models of RCC cells and OBs that mimic the bone tumor microenvironment. In particular, we investigated whether the antitumor activity of cabozantinib persists in the copresence of OBs and whether cabozantinib could directly modulate the OB molecular profile and indirectly modulate the proliferative activity of cancer cells.

## 2. Materials and Methods

### 2.1. Primary Human Osteoblasts

Human primary OBs were obtained from bone marrow samples of healthy patients undergoing total hip replacement at Policlinico Campus Bio-Medico of Rome, Italy. The procedure was approved by the Ethical Committee of the Campus Bio-Medico University of Rome, and informed consent from the patients was collected in accordance with the Declaration of Helsinki principles (Prot 21/15 OSS). Bone marrow mesenchymal stem cells (BM-MSCs) were isolated following the protocol described by Cicione et al. [13]. OB differentiation was achieved by culturing BM-MSCs at an initial density of 5 × 10^4^ in 24-well plates, adding 10-mM beta-glycerophosphate (Sigma-Aldrich, Milan, Italy), 50-μM ascorbic acid (Sigma-Aldrich, Milan, Italy) and 1000 nM dexamethasone (Sigma-Aldrich, Milan, Italy) to the culture medium for 28 days. The OBs were treated or not with cabozantinib at 5 μM from day 21 to day 28.

### 2.2. RCC Cell Lines

Established human RCC cell lines Caki-1, 786-O, Caki-2 and ACHN were purchased from the American Type Culture Collection (ATCC, Manassas, VA, USA). Caki-1 and Caki-2 were grown in McCoy’s 5A medium, 786-O cells in RPMI medium and ACHN cells in Eagle’s Minimum Essential Medium, supplemented with 100 units/liter of penicillin, 100 μg/mL of streptomycin (Euroclone, Pero, Italy), 2 mM of glutamine (Euroclone, Pero, Italy) and 10% fetal bovine serum (Gibco, Waltham, MA, USA) (FBS). Green Fluorescent Protein (GFP)-tagged RCC cell lines were obtained using MISSION^®^ pLKO.1-puro-CMV-Turbo GFP™ Positive Control Transduction Particles (cat. no. SHC003, Sigma-Aldrich, Milan, Italy) with a Multiplicity of Infection (MOI) of 0.5. The transfected cells were then selected, adding 2 μg/mL of puromycin (Euroclone, Pero, Italy).

### 2.3. OBs–RCC Cells “Indirect” Coculture

The osteoblast-conditioned media (OCM) was collected after 48 h of starvation (0.5% of FBS). GFP+ Caki-1 and GFP+ 786-O cells were seeded at 50 × 10^3^ confluency in 24-well plates and treated with OCM for 7 days. The proliferation rate was calculated, measuring the ratio between the GFP signal of the RCC cells after 7 days from seeding and GFP signal after 2 h from the seeding (T0) using Nikon NIS-Elements microscope imaging software.

### 2.4. OBs–RCC Cells “Direct” Coculture

We used two different OBs/RCC tumor cell coculture models, as showed in Figure 1. In model 1, GFP+ RCC cells (50 × 10^3^) were plated on an OB layer in 24-well plates and treated or not with cabozantinib (5 μM) for 7 days (dd); in model 2, GFP+ RCC cells were cultured with OB pretreated or not with cabozantinib (5 μM) for 7 dd. In both coculture models, the GFP signal of the RCC cells was measured after 2 h from the seeding (t0) and after 7 dd using using Nikon NIS-Elements microscope imaging software. The GFP signal at 7 dd was normalized to a GFP signal at t0 and expressed as the proliferation rate. The cabozantinib inhibition rate percentage was calculated as follows: (Ctrl sample − (Cabozantinib/Ctrl)) × 100.

### 2.5. Gene Expression Assay

The total RNA was extracted from OBs treated or not with cabozantinib (5 μM) at the end of the differentiation protocol using a Trizol reagent (Invitrogen, Waltham, MA, USA) according to the manufacturer’s instructions. cDNA was produced using the High-Capacity cDNA Reverse Transcription kit (Applied Biosystems, Waltham, MA, USA) according to the manufacturer’s instructions. The mRNA levels were measured by a quantitative real-time polymerase chain reaction (qRT-PCR) using TaqMan Gene Expression Assays in the 7900 HT Real-Time PCR System (Applied Biosystems, Waltham, MA, USA). The gene expression levels were normalized to the endogenous housekeeping gene Glucuronidase Beta (GUSβ) in both untreated and treated samples using ΔCT calculations. Subsequently, the relative expression levels in the treated samples were normalized to the mRNA levels detected in the control samples using ΔΔCT calculations.

### 2.6. Protein Expression Assay

A proteomic profile analysis was performed on the OBs treated or not with cabozantinib (5 μM). A panel of 507 human target proteins was analyzed using the human antibody Array Membrane Kit (RayBiotech, Peachtree Corners, GA, USA) according to the manufacturer’s instructions. The band signals were detected by the ChemiDoc MTP Imaging System (Bio-Rad, Hercules, CA, USA), and their intensity was quantified using ImageLab software (Bio-Rad, Hercules, CA, USA).

### 2.7. Cell Viability Assay

GFP+ 786-O and GFP+ Caki-1 cells were treated or not with cabozantinib at different doses (10 nM, 100 nM, 500 nM, 1 μM and 5 μM), and after 7 dd, the cell viability was evaluated by a cell growth determination kit with a MTT-based assay (Sigma-Aldrich, Milan, Italy) according to the manufacturer’s instructions.

### 2.8. Statistical Analysis

The data were analyzed using the Student’s *t*-test and One-Way ANOVA, test followed by Tukey’s multiple comparison tests. The D’Agostino-Pearson omnibus normality test was used to check for normal distribution. The graphics processing and statistical tests were performed using the program GraphPad Prism (San Diego, CA, USA).

## 3. Results

### 3.1. Effect of Osteoblasts on Metastatic RCC Cell Proliferation

To evaluate the antiproliferative effects of cabozantinib in a bone microenvironment, we set up a direct cell–cell contact coculture between the OBs and RCC cells. We tested four renal cell lines (786-O and Caki-2 as the primary RCC cells and Caki-1 and ACHN as the metastatic RCC cells), finding that only Caki-1 cells were responsive to osteoblast stimuli (Appendix A). As coculture models, we selected Caki-1 and 786-O cells, which represent the most widespread and best-characterized clear cell RCC lines. These two cell lines showed a different proliferation growth when cocultured with OBs. In particular, Caki-1 cells increased their proliferation rate 2.6-fold in the presence of OBs compared to the cells cultured alone (*p* < 0.0001), while the 786-O cell growth was similar in both culture conditions (Figure 2). These data suggest that OBs could provide fertile soil for metastatic Caki-1 cells, which are more prone to proliferating in response to OB proliferative signals.

To test whether OB cell–cell contact was necessary to stimulate Caki-1 cell growth, we performed an “indirect” coculture treating cells with OB conditioned media. Intriguingly, in this model, we did not observe the same increase in the proliferation of Caki-1 cells observed in the “direct” coculture model (Appendix A), suggesting that the physically interaction with the OBs was needed.

### 3.2. Antitumor Effect of Cabozantinib in RCC Bone Metastatic Models

We investigated if the inhibitory effect of cabozantinib on RCC cell proliferation also persisted in a bone microenvironment (coculture model A). We used a dose of 5 μM, which did not affect the OB viability [12], and it was within the range of the clinical plasma concentrations [14]. In our RCC cell models, this dose represented the half-maximal inhibitory concentration (IC50) in the Caki-1 and 786-O RCC cell lines (Caki-1: Ctrl vs. Cabozantinib 5 μM *p* = 0.011; 786-O: Ctrl vs. Cabozantinib 5 μM; *p* = 0.022) (Figure 3).

The data showed that cabozantinib significantly reduced the proliferation of both Caki-1 (*p* < 0.0001) and 786-O cells (*p* = 0.03) cocultured with OBs. Intriguingly, in Caki-1 cells, the inhibition rate of proliferation was higher in the presence of OBs compared to the monoculture (Figure 4, left panel) (*p* < 0.001); otherwise, the decrease of 786-O cell growth by cabozantinib was similar when the cells were cultured with OBs or alone (Figure 4, right panel).

These results suggest that cabozantinib could exert an additional “indirect” antitumor effect mediated by OBs reducing their ability to boost cancer cell proliferation. To test this hypothesis, Caki-1 and 786-O cells were seeded on OBs previously treated or not with cabozantinib (coculture model B). The data showed that the OB pretreatment with cabozantinib reduced Caki-1 cell growth (*p* = 0.040), confirming its antitumor effect on the OB-mediated cells with metastatic features. The same effect was not observed in the primary 786-O cells (Figure 5).

### 3.3. Effect of Cabozantinib on Osteoblast Gene and Protein Expression

To investigate how cabozantinib modulated the OB molecular profile, we selected a panel of genes known to be involved in tumor cell/OB crosstalk. The data showed that a cabozantinib (5 μM) OB treatment significantly upregulated the expression of jagged canonical notch ligand 1 (JAG1) (*p* = 0.045) while reducing the mRNA levels of other genes, including insulin growth factor (IGF-1) (*p* = 0.033), WNT16 (*p* = 0.032), thrombospondin-1 (THBS1) (*p* = 0.049), secreted protein acidic and cysteine rich (SPARC) (*p* = 0.018), plasminogen activator urokinase (PLAU) (*p* = 0.025) and integrin binding sialoprotein (IBSP) (*p* = 0.038) (Figure 6A).

The reduction of SPARC (*p* = 0.010), THBS-1 (*p* = 0.021) and urokinase-type plasminogen activator (uPA) (*p* = 0.010) expression was also confirmed at the protein level (Figure 6B). In addition, the proteomic analysis revealed the modulation of other signaling molecules in the OBs treated with cabozantinib (Figure 6C). In particular, we found an upregulation of glypican-3 (*p* = 0.015) and ectodysplasin A2 (EDA-A2) (*p* = 0.025) and a reduction of insulin-like growth factor-binding protein 2 (IGFBP-2) (*p* = 0.010), IGFBP-7 (*p* = 0.013) (heregulin 1-apha-/neuregulin 1 (HRG1-alpha/NRG1) (*p* = 0.045), latent TGF-beta binding protein 1 (LTBP1) (*p* = 0.013), monocyte chemoattractant protein-1 (MCP-1) (*p* = 0.011), matrix metalloproteinase-20 (MMP-20) (*p* = 0.046), secreted frizzled-related protein (SFRP-4) (*p* = 0.048), thrombopoietin (*p* = 0.028), tissue inhibitors of metalloproteinase-1 (TIMP-1) (*p* = 0.039), TIMP-2 (*p* = 0.030), vascular endothelial growth factor A (VEGF) (*p* = 0.033) and VEGF-C (*p* = 0.020). All these factors are known to be involved in tumor cell proliferation, and their modulation, after the treatment, could explain the additional antitumor effect of cabozantinib mediated by OBs.

## 4. Discussion

The anticancer efficacy of cabozantinib has been demonstrated in preclinical models of bone metastases from prostate cancer [15,16], as well as its direct impact on osteoclast and OB activity [11,12]. To date, it has not yet been clarified if the antitumor effect of cabozantinib occurs directly on cancer cells and/or indirectly through the modulation of the bone microenvironment.

Here, we used in vitro RCC bone metastatic models to investigate both the “direct” antitumor effect of cabozantinib on the RCC cells and/or its potential “indirect” anticancer activity mediated by OBs.

The choice of two RCC cell lines with different sensitivities to OB stimuli provides the opportunity to evaluate the “real” contribution of OBs in RCC proliferation after a cabozantinib treatment. The different sensitivities of the Caki-1 and 786-O cells to OB signals was, probably, due to their specific molecular and phenotype characteristics. In particular, the metastatic phenotype of Caki-1 cells confers them an increased tendency to interact with the bone microenvironment, whereas the primary 786-O cells could be less responsive to OB factors.

Despite the augmented proliferation of Caki-1 cells on OBs, cabozantinib can completely abrogate their growth, showing an increased inhibitory potency compared to its antitumor effect on cells in a monoculture. Concerning 786-0 cells, the inhibition effect of cabozantinib was similar in both the monoculture and coculture conditions. Intriguingly, the OB pretreatment with cabozantinib affected Caki-1 cell growth, suggesting that the increased antitumor effect of cabozantinib on these cells could be due to its direct effect on OBs. To investigate how a cabozantinib treatment modifies the OB profile, we analyzed the specific genes potentially involved in tumor growth. At the same time, a proteomic analysis of 507 molecules was performed to obtain a more clear and exhaustive picture of OB modulation by cabozantinib. We found some factors significantly inhibited by cabozantinib, both at the gene and protein expression levels, such as Thrombospondin 1, uPA and SPARC. Thrombospondin 1 is a multifaceted player in cancer proliferation, invasion, migration and apoptosis through different interaction networks that are not completely elucidated [17]. uPAR is overexpressed in almost all aggressive malignancies and plays an essential role in promoting tumor progression and metastasis, and its inhibition results in tumor suppression [18]. Regarding SPARC, recently, it has been demonstrated that it is a key mediator of TGF-β-induced renal cancer metastasis and represents a negative prognostic marker of RCC patient survival [19]. Moreover, the extensive proteomic analysis provided us with additional factors downregulated by cabozantinib, some of which could be involved in tumor cell proliferation. Among these, VEGF-A and VEGF-C can directly promote the growth, survival, migration and invasion of cancer cells through VEGF receptor tyrosine kinases (RTKs) [20,21,22]. Moreover, the other two secreted proteins, IGFBP2 and IGFBP7, were reduced in the OBs after the cabozantinib treatment. The evidence showed that the IGFBP2 levels are associated with cancer cell invasion and metastasis [23,24,25]; instead, IGFBP7 displays an ambiguous activity as an oncogene or suppressor gene in distinct types of cancers [26]. A pilot study demonstrated that TIMP-1 is a negative prognostic biomarker in clear cell RCC patients [27]; conversely, EDA-A2 is known to act as tumor suppressor when it binds its receptor, XEDAR [28,29]. Regarding NRG1, it has been recently demonstrated that it is released by the tumor microenvironment and promotes antiandrogen resistance in prostate cancer [30].

Taken together, these data suggest that cabozantinib preserves its efficacy in the in vitro model of RCC bone metastasis and exerts an additional indirect antitumor activity mediated by OBs in metastatic Caki-1 cells. Recently, Pan et al. investigated the effect of cabozantinib in a 3D coculture model of bone metastatic renal cells and a pre-OB cell line to mimic the tumor/bone interplay. In this model, cabozantinib was able to increase the OB number and bone volume, reverting the osteoblast inhibition mediated by the RCC cells [31]. This anabolic effect of cabozantinib reported by the authors could result in the modulation of OB protein secretion that we observed in our study. Moreover, our data are in accordance with the results observed in the METEOR study [9] that reported a significant clinical benefit in the bone metastatic RCC patients subset.

Not identifying a single factor or specific pathway responsible for the antitumor effect of cabozantinib mediated by OBs seems to be a limitation of the study. However, we think that the antitumor activity of cabozantinib could be mediated by different OB molecules rather than a single one. Although this study is an explorative analysis, it could provide biological information about cabozantinib activity with a potential clinical utility. The use of RCC in vitro models undoubtedly represents an important limitation, but, on the other hand, the cell–cell contact coculture system based on primary human OBs could resemble the in vivo tumor bone microenvironment. The complete elucidation of the OB contribution in RCC cell proliferation could be useful to design novel and more effective anticancer combination strategies to improve the efficacy of the existing treatments [32].

## 5. Conclusions

Although OBs are important components of the bone metastatic niche, their role in skeletal metastases development has been relatively under-investigated. Our study identified some specific OB signaling molecules that could support tumor cell proliferation and survival within bone.

From a clinical perspective, the elucidation of the molecular mechanisms behind tumor cell/OB interactions may be essential in identifying novel therapeutic strategies to inhibit cancer progression in bone. In addition, our in vitro cell–cell coculture models could provide simple but useful systems to investigate the tumor cell/OB crosstalk. Finally, it could provide a suitable platform to evaluate the direct effect of different anti-cancer agents on the bone tumor microenvironment.

## Figures and Tables

**Figure 1 biology-10-00781-f001:**
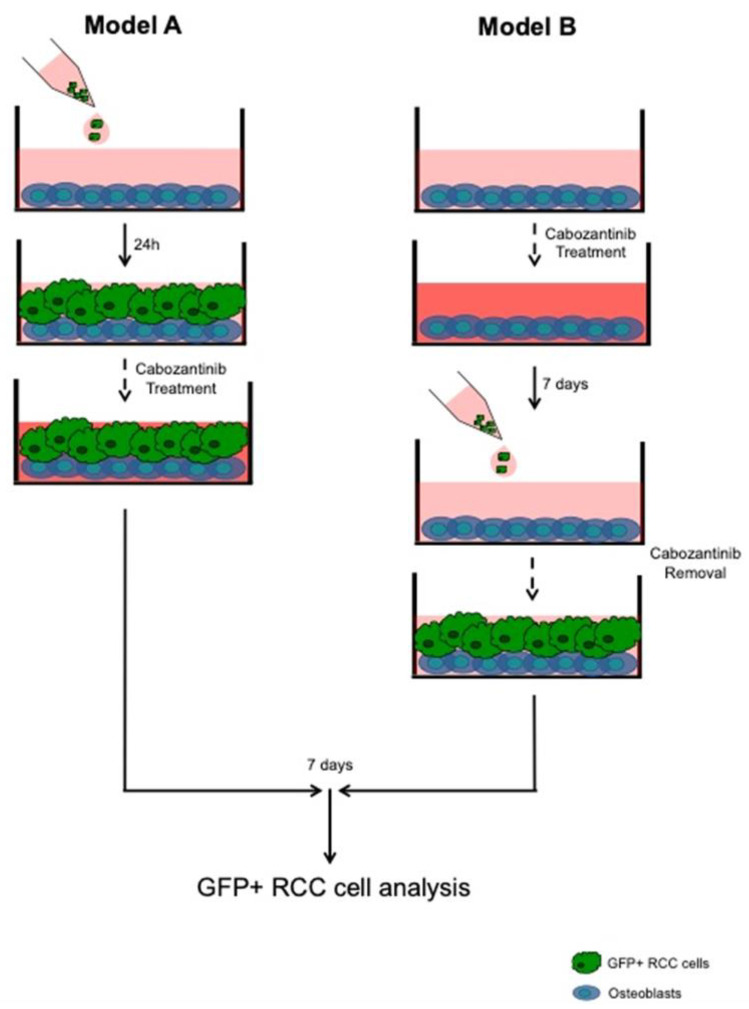
Representative schemes of OBs/RCC tumor cell coculture models. Model (**A**): 786-O GFP+ cells and GFP+ Caki-1 cells were seeded on an OB layer and treated or not with cabozantinib at 5 μM. After 7 dd, the GFP signal of the RCC cells was measured. Model (**B**): The OBs were treated or not with cabozantinib at 5 μM for 7 dd. GFP+ RCC cells were added, and the GFP signal was measured after 7 dd.

**Figure 2 biology-10-00781-f002:**
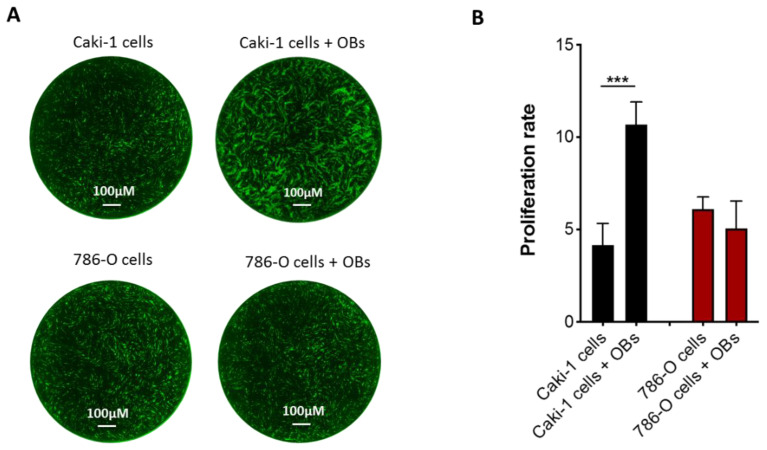
Effects of OBs on metastatic RCC cell proliferation. (**A**) Representative images of Caki-1 GFP+ (green) and 786-O GFP+ cells (green) cultured alone or with OBs. Scale bar: 100 μM and magnification 4×. (**B**) Proliferation rate analysis of Caki-1 GFP+ and 786-O GFP+ cells in a monoculture or cocultured with OBs. Data are expressed as the mean ± SD; *** *p* < 0.001.

**Figure 3 biology-10-00781-f003:**
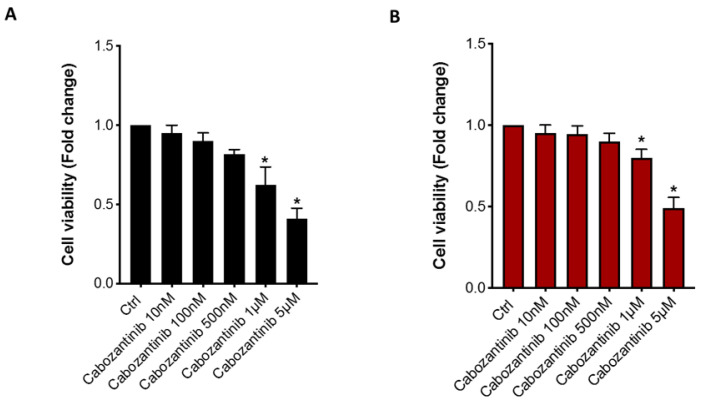
Antitumor effect of cabozantinib in RCC cells. Viability assay (MTT) of Caki-1 (**A**) and 786-O cells (**B**) after treatments with different doses of cabozantinib (10 nM, 100 nM, 500 nM, 1 μM and 5 μM). Data are expressed as the mean ± SD; * *p* ≤ 0.05.

**Figure 4 biology-10-00781-f004:**
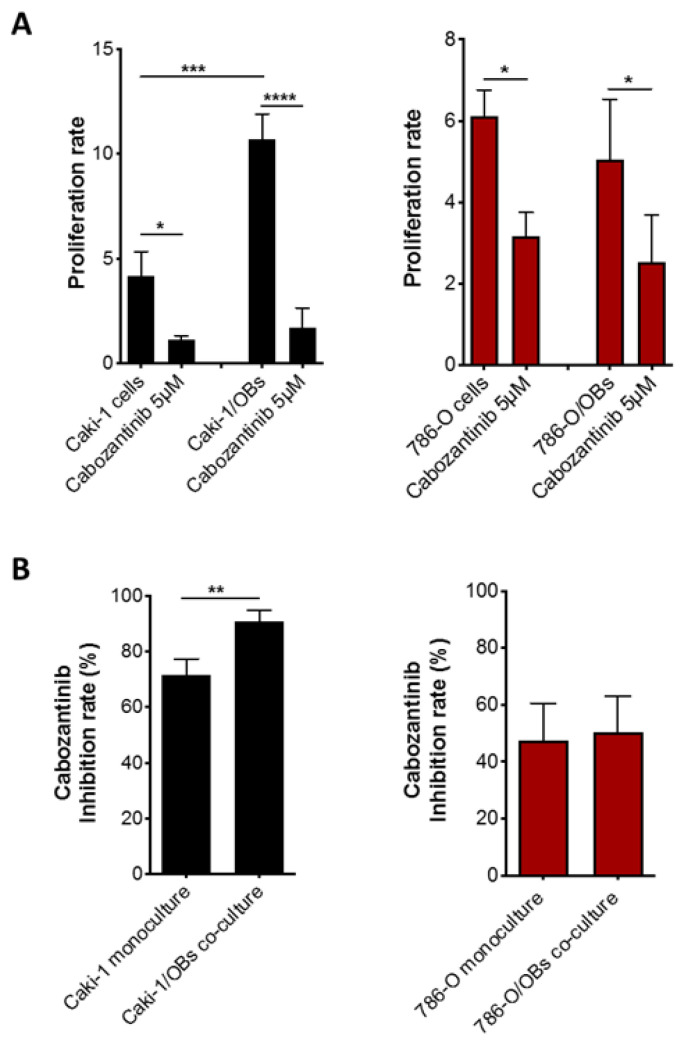
Antitumor effect of cabozantinib in the RCC bone microenvironment. Proliferation (**A**) and inhibition rates (**B**) of Caki-1 GFP+ (left panels) and 786-O GFP+ (right panels) cells after a cabozantinib treatment (5 μM) in a monoculture or cocultured with OBs. Data are expressed as the mean ± SD; * *p* ≤ 0.05, ** *p* < 0.01, *** *p* < 0.001 and **** *p* < 0.0001.

**Figure 5 biology-10-00781-f005:**
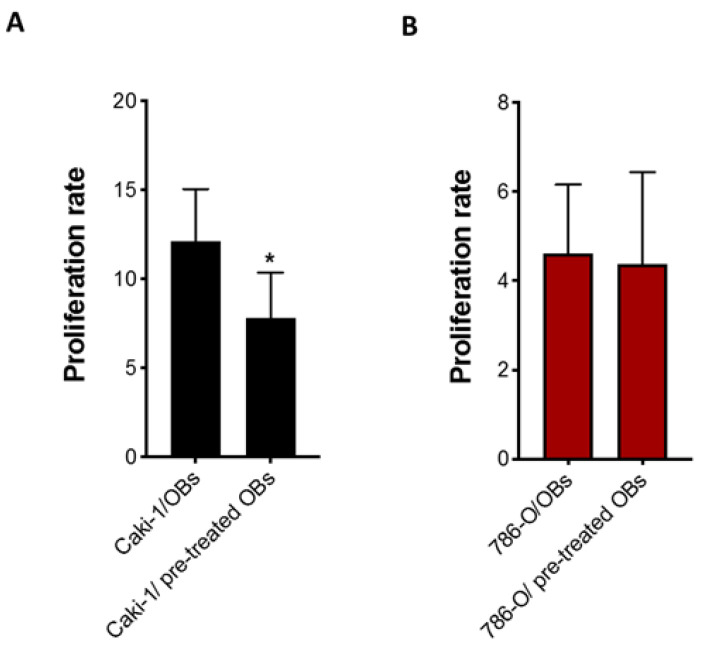
The antitumor effect of cabozantinib mediated by OBs. The proliferation rate of Caki-1 GFP+ (**A**) and 786-O GFP+ (**B**) cells cultured with OBs pretreated with cabozantinib (5 μM). Data are expressed as the mean ± SD; * *p* ≤ 0.05.

**Figure 6 biology-10-00781-f006:**
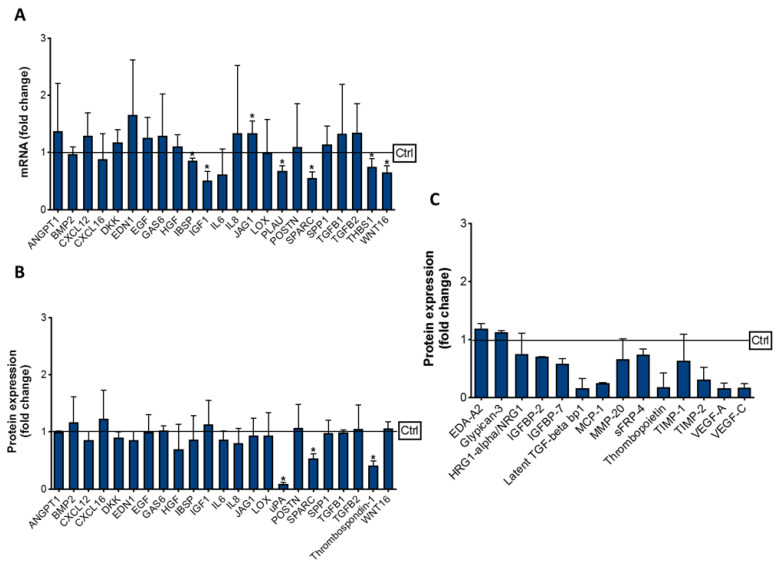
Effect of cabozantinib on the OB molecular profile. The OB gene (**A**) and protein expression (**B**) of selected factors after the cabozantinib treatment (5 μM). (**C**) Other molecules significantly modulated by the cabozantinib treatment. Data are presented as the fold change with respect to the untreated OBs (Ctrl = 1). Data are presented as the mean ± SD; * *p* ≤ 0.05.

## Data Availability

Not applicable.

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
