# Peer review of "Antitumor Effect of Cabozantinib in Bone Metastatic Models of Renal Cell Carcinoma"

_biology, 2021, doi:10.3390/biology10080781_

Round 1
Reviewer 1 Report
1. Regarding the background and literature review, there was a recent article published (Pan et. al., Mol Cancer Ther, 2020, 19(6):1266) that looks at cabozantinib in renal cell carcinoma bone mets in vitro and in vivo. This group also used 786-O cells, and a bone specific cell line derived from 786-O, in co-culture with osteoblasts. Although I did not see proliferation data, cabozantinib effects on osteoblasts were examined. The authors of the current manuscript should review and comment.
2. Regarding the experimental design, the findings were with one cell line, Caki-1. It would be suggested to test another cell line that is also responsive to osteoblast co-culture.
3. In addition, the authors only used 2D co-culture with cell-cell contact. As the cabozantinib effect seems to also be via osteoblasts, it would be good to test whether cell-cell contact is necessary. It would be suggested to us a non-contact co-culture system (i.e. boyden chamber) to evaluate this.
4. The other limitation is 2-D culture. Using 3-D culture also would add insight as phenotypes often are responsive to the more physiologic environment provided by 3D.
5. Finally, some in vivo validation would be suggested. At minimum, showing that the caki-1 cells form bone metastasis in a mouse model would be suggested.
Author Response
- Regarding the background and literature review, there was a recent article published (Pan et. al., Mol Cancer Ther, 2020, 19(6):1266) that looks at cabozantinib in renal cell carcinoma bone mets in vitro and in vivo. This group also used 786-O cells, and a bone specific cell line derived from 786-O, in co-culture with osteoblasts. Although I did not see proliferation data, cabozantinib effects on osteoblasts were examined. The authors of the current manuscript should review and comment.
We provided to discuss this manuscript in the “discussion section”.
- Regarding the experimental design, the findings were with one cell line, Caki-1. It would be suggested to test another cell line that is also responsive to osteoblast co-culture.
Actually, we tested four renal cell lines (786-O and Caki-2 as primary RCC, Caki-1 and ACHN as metastatic RCC) finding that only caki-1 cells were responsive to osteoblast stimuli (see attached figure 1). Thus, we decide to proceed with cabozantinib treatment of more widespread and best-characterized primary and metastatic RCC lines, 786-0 and caki-1. In addition, 786-0 and caki-1 have the same histological RCC clear cell subtype.
- In addition, the authors only used 2D co-culture with cell-cell contact. As the cabozantinib effect seems to also be via osteoblasts, it would be good to test whether cell-cell contact is necessary. It would be suggested to us a non-contact co-culture system (i.e. boyden chamber) to evaluate this.
This is a good point. At the beginning of study, we performed, at the same time, a “direct” cell-cell contact co-culture OB/RCC cells and an “indirect” co-culture treating RCC cells with OB supernatant. Intriguingly we found that in direct co-culture caki-1 cells increased their proliferation in presence of OBs, while in indirect co-colture we did not observe the same phenomenon (see attached figure 2). These data suggest that the physically presence of OBs provide proliferative signals to caki-1 cells influenced their growth. For these reason, we decided to test the effect of cabozantinib only on a direct cell-cell contact co-culture that better resemble tumor bone microenvironment and OB/RCC cells crosstalk.
- The other limitation is 2-D culture. Using 3-D culture also would add insight as phenotypes often are responsive to the more physiologic environment provided by 3D.
We are absolutely agree with reviewer that 3-D culture systems better resemble the complexity of bone/tumor interactions, but our in vitro model of primary human osteoblasts are highly reliable as documented by our previous publications:
Iuliani M, Pantano F, Buttigliero C, Fioramonti M, Bertaglia V, Vincenzi B, et al. Biological and clinical effects of abiraterone on anti-resorptive and anabolic activity in bone microenvironment. Oncotarget. 2015 May 20;6(14):12520-8. doi: 10.18632/oncotarget.3724. PMID: 25904051; PMCID: PMC4494955.
Fioramonti M, Santini D, Iuliani M, Ribelli G, Manca P, Papapietro N, et al. Cabozan-tinib targets bone microenvironment modulating human osteoclast and osteoblast functions. Oncotarget. 2017;8:20113
Fioramonti M, Fausti V, Pantano F, Iuliani M, Ribelli G, Lotti F, et al. Cabozantinib Af-fects Osteosarcoma Growth Through A Direct Effect On Tumor Cells and Modifications In Bone Microenvironment. Sci Rep. 2018;8:4177
Moreover, 3-D culture models of bone tumor microenvironment mainly use murine osteoblast cell lines (i.e. MC3T3) (Pan et. al., Mol Cancer Ther, 2020, 19(6):1266) with reduced time of growth and different culture conditions compared to more physiologic primary human models.
- Finally, some in vivo validation would be suggested. At minimum, showing that the caki-1 cells form bone metastasis in a mouse model would be suggested.
This is another good point, but, unfortunately, no data are available in literature about Caki-1 ability to metastasize to the bone in murine models. Nevertheless, Caki-1 cells were selected because of metastatic origin and, in our experiments, showed an enhanced growth on osteoblast layer compared to the plastic suggesting that Caki-1 proliferation may be influenced by osteoblasts. Based on these results, we believe that caki-1 are a good model to study osteoblast and tumor cell interaction and the effect of anticancer treatments.

Reviewer 2 Report
In this article, the authors investigated whether the anti-tumor activity of cabozantinib persists in the co-presence of osteoblasts and whether cabozantinib could directly modulate osteoblast molecular profile and indirectly the proliferative activity of cancer cells. The manuscript is straightforward, well written, concise and has clear results. Definitely deserves to be published and is a valuable contribution to the “biology” journal. Some minor flaws need to be addressed before publication.
Minor points:
[1] “1. Introduction”, Page 2 of 12, Lines 56-57:
“The most common metastases in renal cell carcinoma (RCC) occur to the lung, followed by bone involvement in 20–35% [1-2].”.
Which are the other metastatic sites, according the natural history of RCC? Please, report that the RCC being the most common primary cancer site resulting in an isolated pancreatic metastasis, which can occur a long time period after nephrectomy; this spread could occur as late as 10–32 years.
[2] “1. Introduction”, Page 2 of 12, Lines 67-68:
“Recently, US-FDA extended its indication to all advanced RCC patients.”.
At that stage, please, make a comment about the safety of cabozantinib. Clinicians should be aware of infrequent or serious events that can occur, specifically in the later phases of treatment with cabozantinib. These include hemorrhagic events of grade 3 or higher that may arise as a result of reduced vascular integrity, along with arterial and venous or mixed thrombotic events. Data on the long-term risk-benefit profile of the direct oral anticoagulants for cabozantinib-associated thrombosis will be monumental for future clinical practice and reduce morbidity and mortality in this patient population.
Recommended reference: Shah S, et al. Cancer-Associated Thrombosis: A New Light on an Old Story. Diseases. 2021;9(2):34.
[3] Does the “RCC” include all the histological subtypes or it is just the clear cell? Please, clarify accordingly.
[4] Please, add a “5. Conclusion” section.
Author Response
Minor points:
[1] “1. Introduction”, Page 2 of 12, Lines 56-57:
“The most common metastases in renal cell carcinoma (RCC) occur to the lung, followed by bone involvement in 20–35% [1-2].”.
Which are the other metastatic sites, according the natural history of RCC? Please, report that the RCC being the most common primary cancer site resulting in an isolated pancreatic metastasis, which can occur a long time period after nephrectomy; this spread could occur as late as 10–32 years.
We included this information as suggested.
[2] “1. Introduction”, Page 2 of 12, Lines 67-68:
“Recently, US-FDA extended its indication to all advanced RCC patients.”.
At that stage, please, make a comment about the safety of cabozantinib. Clinicians should be aware of infrequent or serious events that can occur, specifically in the later phases of treatment with cabozantinib. These include hemorrhagic events of grade 3 or higher that may arise as a result of reduced vascular integrity, along with arterial and venous or mixed thrombotic events. Data on the long-term risk-benefit profile of the direct oral anticoagulants for cabozantinib-associated thrombosis will be monumental for future clinical practice and reduce morbidity and mortality in this patient population.
Recommended reference: Shah S, et al. Cancer-Associated Thrombosis: A New Light on an Old Story. Diseases. 2021;9(2):34.
We included a brief comment about the safety of cabozantinib.
[3] Does the “RCC” include all the histological subtypes or it is just the clear cell? Please, clarify accordingly.
We provided to clarify this point in the introduction section.
[4] Please, add a “5. Conclusion” section.
We added a conclusion section.
Reviewer 3 Report
The manuscript entitled ‘Anti-tumor effect of cabozantinib in bone metastatic model of renal cell carcinoma’ by Iuliani et al. is very interesting and suitable for publications. However, to improve the quality of manuscript author need to address the following comments
Introduction is too short; author need to elaborate with background and current strategies have been developed and challenges remain.
Authors need to highlight the major focus of the manuscript in abstract, conclusion, and introduction
Quality of the figure 1 is poor, author need to look at literature for proper presentation of cells in dish
The text font size needs to be improved in figure 2
Authors need to cite the following literature
https://doi.org/10.1039/D0TB01559H
abbreviation must be disclosed at their first appearance, author need to take care of them carefully. For eg. US-FDA and so on
Author Response
Introduction is too short; author need to elaborate with background and current strategies have been developed and challenges remain.
We provided to deepen the background section underlining the current evidences and open challenges.
Authors need to highlight the major focus of the manuscript in abstract, conclusion, and introduction
We provided to highlight the focus and the rationale of the study in the abstract and introduction section. Moreover, we added a conclusion section.
Quality of the figure 1 is poor, author need to look at literature for proper presentation of cells in dish
We modified figure 1 according to the most common schematic representations of cell culture of published papers.
The text font size needs to be improved in figure 2
We provided to increase font size in figure 2 to improve it.
Authors need to cite the following literature
https://doi.org/10.1039/D0TB01559H
We included this reference.
abbreviation must be disclosed at their first appearance, author need to take care of them carefully. For eg. US-FDA and so on
We revised all abbreviations in the text.